# Echocardiography Nomogram for Predicting Survival among Chronic Lung Disease Patients with Severe Pulmonary Hypertension

**DOI:** 10.3390/jcm11061603

**Published:** 2022-03-14

**Authors:** Rong Jiang, Lan Wang, Qin-Hua Zhao, Cheng Wu, Ping Yuan, Shang Wang, Rui Zhang, Su-Gang Gong, Wen-Hui Wu, Jing He, Hong-Ling Qiu, Ci-Jun Luo, Jin-Ming Liu, Zhi-Cheng Jing

**Affiliations:** 1Department of Cardiopulmonary Circulation, Shanghai Pulmonary Hospital, Tongji University School of Medicine, Shanghai 200433, China; listening39@tongji.edu.cn (R.J.); lanwang@tongji.edu.cn (L.W.); zhaoqinhua@tongji.edu.cn (Q.-H.Z.); pandyyuan@tongji.edu.cn (P.Y.); 2111697@tongji.edu.cn (S.W.); zhangrui@tongji.edu.cn (R.Z.); gongsugang@tongji.edu.cn (S.-G.G.); wenhui_wu@tongji.edu.cn (W.-H.W.); 1801230@tongji.edu.cn (J.H.); qiuhongling@tongji.edu.cn (H.-L.Q.); luocj@tongji.edu.cn (C.-J.L.); jinmingliu@tongji.edu.cn (J.-M.L.); 2Department of Health Statistics, Naval Medical University, 800 Xiangyin Road, Shanghai 200433, China; wucheng@smmu.edu.cn; 3Department of Cardiology, Peking Union Medical College Hospital, Chinese Academy of Medical Sciences, 1 Shuai-Fu-Yuan, Dongcheng District, Beijing 100730, China

**Keywords:** chronic lung diseases, echocardiography, pulmonary hypertension, haemodynamics, survival, right heart catheterization

## Abstract

Severe pulmonary hypertension in chronic lung diseases (severe CLD-PH) differs significantly from other types of PH in physiology and prognosis. We aimed to assess whether echocardiography helps predict long-term survival in patients with severe CLD-PH. This single-centre, observational cohort study enrolled 100 patients with severe CLD-PH (mean pulmonary arterial pressure ≥35 mm Hg or ≥25 mm Hg with cardiac index <2.0 L/min/m^2^ or pulmonary vascular resistance ≥6 Wood units) between 2009 and 2014. The population was randomly divided into a derivation and validation cohort in a 2:1 ratio. To construct a nomogram, a multivariable logistic regression model was applied, and scores were assigned based on the hazard ratio of independent echocardiographic predictors. Multivariate Cox hazards analysis identified the strongest predictors of mortality as pulmonary arterial systolic pressure (PASP), tricuspid annular plane systolic excursion, and right ventricular end-diastolic transverse dimension. The three independent predictors were entered into the nomogram. Compared with PASP alone, the nomogram resulted in an integrated discrimination improvement of 15.5% (95% confidence interval, 5.52–25.5%, *p* = 0.002) with a net improvement in model discrimination (C-statistic from 0.591 to 0.746). Using echocardiographic parameters, we established and validated a novel nomogram to predict all-cause death for patients with severe CLD-PH.

## 1. Introduction

Pulmonary hypertension due to chronic lung diseases (CLD-PH) is classified as World Health Organization (WHO) Group 3, per the classification of the World Symposium on PH [1,2,3]. According to right heart catheterization (RHC), PH in CLD is classified as without/mild PH or severe CLD-PH, where severe CLD-PH is defined as a mean pulmonary artery pressure (mPAP) ≥35 mm Hg or ≥25 mm Hg with a low cardiac index (CI) (<2 L/min/m^2^) [2]. Severe CLD-PH accounts for a minority of patients with CLD and seems to be characterized by extensive general vascular remodelling and a poor prognosis [4,5]. As these patients have the “pulmonary vascular phenotype”, pulmonary arterial hypertension (PAH)-targeted therapy may be considered for the treatment of severe CLD-PH. It is important to predict the survival of severe CLD-PH patients because the most at-risk patients could be considered for lung transplantation if eligible.

Echocardiography is considered to be the best non-invasive method for screening and evaluating PH. Because they provide accurate information on cardiac structure and physiology, echocardiographic-derived multivariable risk models can reliably predict survival in WHO Group 1 PH patients [6,7,8,9]. However, the ability to estimate pulmonary artery systolic pressure (PASP) by assessing peak tricuspid regurgitation velocity has been limited in CLD patients [10,11]. Per the 6th World Symposium on PH [12], alternative PH signs such as ventricular septal shift, right ventricle (RV) enlargement, pulmonary artery dilation, impaired RV function, etc., were predictors of unfavourable outcomes and were routinely used for risk stratification in WHO Group 1 [13,14]. In our previous study, we established a novel scoring index with guideline-recommended echocardiographic parameters to predict severe CLD-PH [15]. Whether echocardiographic parameters can predict mortality in severe CLD-PH remains unknown.

Our study aimed to explore the prognostic value of guideline-recommended echocardiographic parameters for all-cause mortality in patients with severe CLD-PH.

## 2. Materials and Methods

### 2.1. Patient Selection and Study Design

This was a retrospective study. We reviewed incident patients with suspected PH associated with CLD referred to our centre between 2009 and 2014. All patients underwent Doppler echocardiography and RHC for haemodynamic evaluation. For the haemodynamic characteristics, patients were classified into 3 subsets [2,3]: without PH (mPAP < 25 mmHg); mild PH (25 ≤ mPAP < 35 mmHg); and severe PH (mPAP ≥ 35 mmHg or 25 mmHg ≤ mPAP < 35 mmHg with CI < 2.0 L/min/m^2^ or pulmonary vascular resistance [PVR] > 6 Wood units). The inclusion criteria were (1) diagnosis of CLD confirmed by experienced specialists according to the appropriate guidelines [16,17]; (2) severe PH [1,2,3]; and (3) echocardiography and RHC performed after patients received optimal medical treatment. The exclusion criteria were (1) CLD patients with mild PH or without PH; (2) patients in other WHO PH groups on the basis of the NICE criteria [1]; (3) no RHC or echocardiography at a clinically stable stage; or (4) other severe comorbidities, such as severe left heart diseases and pulmonary embolism.

Patients were randomly assigned to either the derivation or validation cohort in a 2:1 ratio by a permuted randomization protocol after they were diagnosed with severe CLD-PH. Risk factor analysis and nomogram development were created in the derivation cohort and verified in the validation cohort.

This study was approved by the ethics committee of Shanghai Pulmonary Hospital (K08-015C) and conducted in accordance with the amended Declaration of Helsinki. All participants provided written informed consent.

### 2.2. Echocardiographic Measurement

The process of transthoracic echocardiography was described in our previously published article [15,18]. Guideline-recommended additional PH sign parameters included right ventricular end-diastolic transverse dimension (RVEDTD), right ventricular end-diastolic longitudinal dimension (RVEDLD), right atrial transverse dimension (RATD), right atrial longitudinal dimension (RALD), pulmonary artery dimension (PAd) and ventricular septal shift reflected by the end-systolic stage left ventricular eccentricity index (ENDSEI). PASP was measured by peak tricuspid regurgitation velocity combined with right atrial pressure (RAP), which was estimated by inferior cava diameter and inspiratory collapse. RV function, which was reflected by tricuspid annular plane systolic excursion (TAPSE) and peak velocities of the lateral tricuspid annulus (TV s’), was calculated from tissue Doppler images. Left ventricular cardiac structure and function, including the left ventricular end-diastolic transverse dimension (LVDED) and left ventricular end-systolic transverse dimension (LVSED), were also measured. The two cardiologists who performed the echocardiography were blinded to the patients’ information.

### 2.3. Haemodynamic Measurements

RHC was performed by using standard techniques with haemodynamic measurements obtained at rest [19]. The haemodynamic variables included mean RAP (mRAP), mPAP, pulmonary artery wedge pressure (PAWP), cardiac output (CO), CI, and PVR.

#### Follow-up

Clinical, demographic, echocardiographic and RHC parameters were collected from hospital records. Mortality information during follow-up was obtained from chart review, outpatient clinic visits or telephone interviews. The follow-up period lasted from the date of echocardiography until the patient was censored or died at the end of the study (31 June 2017). The primary endpoint was all-cause mortality.

### 2.4. Statistical Analysis

The baseline characteristics of the two cohorts are described as the mean ± standard deviation, median (25th–75th percentile), or counts (proportions) for normally distributed variables, variables with a skewed distribution and categorical variables, respectively. The optimal cut-off values of the parameters from log-rank χ^2^ statistics were identified by the X-tile program (http://www.tissuearray.org/rimmlab/, accessed on 31 March 2018) and divided into low- and high-risk subsets [20].

Univariate and multivariate Cox proportional hazards regression analyses were applied to assess the relations between all-cause mortality and the covariates of interest. The variables that were significant in univariate analysis were included in the multivariate Cox regression model analysis. Independent predictors of all-cause mortality were identified with a p-to-enter of 0.10 or less and a p-to-remove of 0.15 or more. Integer scores distributed on the basis of multiples of rounded β-coefficients from the refitted model were applied to create a new composite scoring system. The composite scoring system was then used for each patient and included in the Cox multivariate analysis for all-cause mortality prediction. Wald chi-square statistics and 95 confidence intervals (CIs) were used to calculate the significance of the estimates at a level of 0.05. The discriminatory ability of scores to predict all-cause death was assessed by receiver operating characteristic (ROC) curves, and sensitivity and specificity were calculated. Event-free survival was estimated from the time of echocardiography, with all-cause mortality as the endpoint. Kaplan–Meier survival curves and log-rank tests identified scores for survival outcomes. A nomogram for significant parameters associated with 1-, 3-, and 5-year survival was generated.

Statistical analysis was performed using R software (version 4.05; http://www.r-project.org, accessed on 31 March 2021) and SPSS 21.0 software (SPSS Inc., Chicago, IL, USA). In all univariate analyses, *p* < 0.05 was considered indicative of statistical significance.

## 3. Results

### 3.1. Study Population

A total of 2251 patients with CLD from May 2009 to October 2014 were entered into our centre database (Figure 1). Finally, severe CLD-PH patients were enrolled and randomly divided into a derivation (*n* = 67) or validation (*n* = 33) cohort in a 2:1 ratio after ineligible patients were excluded (Figure 1). A comparison of the demographics and baseline characteristics was made between the severe CLD-PH patients in the derivation cohort and those in the validation cohort (Table 1). Furthermore, Appendix A shows the comparison of lung function tests in the derivation cohort, stratified by obstructive or restrictive pattern, mixed restrictive/obstructive pattern, and hypoxia without lung disease.

### 3.2. Risk Score Construction in the Derivation Cohort

The majority of patients were male (56.7%), and there was a large proportion of chronic obstructive pulmonary disease (COPD) in the cohort (77.6%), as shown by a mPAP of 46.0 mmHg, CI of 3.1 L/min/m^2^ and PVR of 8.0 Wood units (Table 1). When first admitted to our centre, most patients were initiated on bronchodilator treatment and oxygen- and PAH-targeted drugs, including prostacyclin analogues, phosphodiesterase-5 inhibitors and endothelin receptor antagonists. All-cause mortality occurred in 40 (61.2%) patients during a median follow-up of 2.4 (0.8, 5.4) years. The follow-up rate was 100%. Most patients discontinued PAH-targeted drugs after 3 to 6 months but with the long-term sustainability of respiratory support therapy, such as bronchodilators. The survival rates were 68.7% (CI, 57.5–79.9%) at 1 year, 49.3% (CI, 37.2–61.2%) at 3 years, and 8.9% (CI, 19.1–46.1%) at 5 years (Figure 2A).

Overall, 80.6% of patients had an enlarged right atrium (RATD ≥ 4.4 cm or RALD ≥ 5.1 cm), while 83.6% of patients had an enlarged RV, namely, RVEDTD ≥ 3.5 cm or RVEDLD ≥ 8.6 cm (Table 1). TAPSE < 20 mm was present in 77.6% of patients, while only 25.4% had a TAPSE < 15 mm, suggesting mildly impaired RV in most patients. Furthermore, while TV s’ < 12 cm/s was present in 34.3% of patients, it was below 8 cm/sec in only 7.4% of patients. The PASP was measured at a clinically stable stage in 94.0% of all patients. The degree of left ventricular compression by the RV, defined as ENDSEI > 1.0, was present in 74.6% of patients. Overall, pulmonary artery dilation, defined as a PAd ≥ 2.7 cm, was present in 70.1% of patients, and nearly half (43.2%) had a PAd ≥ 3.0 cm.

The independent prognostic parameters by univariate Cox regression analysis were LVSED, RVEDTD, PASP, TAPSE, ENDSEI, forced expiratory volume in 1 s (FEV1)/forced vital capacity (FVC) and diffusing capacity for carbon monoxide (DLco) predicted %. The high- and low-risk subgroups were divided in terms of 5-year all-cause mortality by the X-tile program (Figure 3). Then, the above parameters were transformed into binary variables based on the cut-offs by the X-tile software.

Moreover, based on clinical significance and significant differences, LVSED, RVEDTD, PASP, TAPSE and ENDSEI were selected for multivariate Cox regression. As predictors of all-cause mortality, multivariate Cox regression identified TAPSE, RVEDTD, and PASP, with cut-offs of 1.8 cm, 4.2 cm and 103 mmHg, respectively. In the multivariable analysis, TAPSE (HR 0.414, 95% CI 0.196–0.873, *p* = 0.021), RVEDTD (HR 2.248, 95% CI 1.062–4.759, *p* = 0.034) and PASP (HR 5.039, 95% CI 2.002–12.680, *p* = 0.001) were associated with all-cause death (Table 2). Integer scores were assigned values of −1, 1 and 2 for the β-coefficient associated with HRs of −0.882, 0.810 and 1.617, respectively. A composite score based on the strongest β-coefficients in the final model (Table 3) was generated: one point was attributed to RVEDTD ≥ 4.2 cm, one point was attributed to PASP ≥ 103 mm Hg and minus one point was attributed to TAPSE ≥ 1.8 cm.

PASP ≥ 103 mm Hg displayed 22.0% sensitivity and 96.2% specificity [area under the curve (AUC): 0.591, 95% CI: 0.464–0.709, *p* = 0.214] by ROC analysis. Compared with PASP, the composite score resulted in an integrated discrimination improvement of 15.5% (95% CI: 5.52–25.5%, *p* = 0.002) and a net improvement in model discrimination with a C-statistic from 0.591 to 0.746 (Figure 4).

In Kaplan–Meier analysis, significant differences in survival were observed among patients with scores of −1, 0, 1, 2 and 3 (log rank = 14.756, *p* = 0.00015; Figure 2B). Compared with severe CLD-PH patients with a score ≥ 0, those with a score < 0 had a higher survival rate (estimated 5-year survival of 13.2% vs. 3.4%; *p* < 0.0001) (Figure 2C).

### 3.3. Nomogram for Predicting All-Cause Mortality

We used a nomogram on the basis of the results of the multivariate analyses to predict 5-year overall survival rates. PASP, RVEDTD and TAPSE levels were included for the cohort (Figure 5). The predictive models showed that high RVEDTD and PASP were adverse prognostic factors, while a high TAPSE level was a favourable factor.

### 3.4. External Validation in the Validation Cohort

In the validation cohort, all-cause mortality occurred in 24 (72.7%) patients, and the nomogram had a C index of 0.737 (95% CI 0.526–1.474, *p* < 0.0001). There were also perfect calibration curves for all-cause death according to the mean of the total points calculated by summing each patient’s nomogram (mean survival of 3.84 vs. 2.74 years, high risk vs. low risk, respectively, *p* = 0.0042) (Figure 2D).

### 3.5. Subgroup Analysis

We analysed 52 COPD patients in the derivation cohort, and all-cause mortality occurred in 33 patients. Lactic acid, FVC, FEV1/FVC, DLco, LVSED, RVEDTD, PASP, TAPSE and ENDSEI were selected for multivariate Cox regression (Table 4). The predictor of all-cause mortality was RVEDTD, with a cut-off of 4.1 cm. The risk of all-cause death increased by 5.65 (95% CI: 1.529–20.9, *p* = 0.009) with an RVEDTD ≥ 4.1 cm by Cox regression analysis (Figure 6).

## 4. Discussion

The association of specific echocardiographic parameters associated with clinical outcomes has been investigated in multiple PAH studies [6,14,21,22,23,24]. However, it is relatively unclear which particular metrics are most suitable in CLD patients with severe PH or add information to risk scores. Our study aimed to compare and evaluate the relationships between multiple echocardiographic parameters and all-cause mortality in patients with severe CLD-PH. We revealed that the new composite echocardiographic scores had a good ability to predict all-cause mortality in severe CLD-PH. The nomogram may be applied for risk stratification of patients with severe CLD-PH.

Our data confirm significant differences between severe CLD-PH and other types of PAH. In 2007, we reported a Chinese registry and survival study of idiopathic PAH (IPAH) [25]. Similar to IPAH patients in the US National Institutes of Health registry in the 1980s, i.e., the traditional management era, the 1- and 3-year survival rates were only 68% and 39%, respectively, in China [26]. In the modern management era, the 1-, 3- and 5-year survival rates of IPAH/familial PAH patients were 85%, 68% and 57% and 85.7%, 69.6% and 54.9% in the REVEAL Registry [7,27] and the French Registry [28], respectively. PAH patients in China benefit from targeted therapies, and the 1- and 3-year survival rates were 92.1% and 75.1%, respectively, in patients with IPAH and 85.4% and 53.6%, respectively, in those with PAH associated with connective tissue diseases (CTD-PAH) [29]. However, our study revealed that patients with severe CLD-PH had a much poorer prognosis than IPAH and CTD-PAH patients, even in the traditional management era.

In contrast to Andersen et al. [30], who studied mild PH related to end-stage COPD, we recognized a significant adverse effect on 5-year survival of severe CLD-PH in our present study (8.9% vs. 37%). Interestingly, 1-year survival rates were lower in the present study group than in those with mild PH in advanced idiopathic pulmonary fibrosis (IPF) (68.7% vs. 72%) [31]. Similar to the characteristics of IPAH, morphologic lesions can be observed in PH associated with COPD that develop to levels characteristic of IPAH [32]. The extent of pulmonary vascular lesions in COPD was correlated with the severity of PH [32]. Compared with CLD patients without or with mild PH, those with severe PH exhibited higher mortality [13,33]. Therefore, it is essential to detect severe PH earlier and to initiate optimal therapy, such as lung transplantation, to improve patient prognosis.

RV function was associated with prognosis, and the longitudinal function of the RV, reflected by the TAPSE, was maintained or mildly impaired and associated with poor prognosis in our study. Although other known echocardiographic parameters, such as EI and pericardial effusion [14], showed prognostic value in IPAH, ENDSEI and pericardial effusion did not predict outcomes in our cohort. Nevertheless, the LVSED, which also represents the degree of LV compression, appears to be predictive beyond the ENDSEI. Pulmonary artery dilatation was common in our severe CLD-PH cohort and present in nearly half of the patients; however, it did not have prognostic value for all-cause death.

Compared with that of IPAH patients, the long-term prognosis of severe CLD-PH patients was far from normal due to more aggressive disease progression. RV dilation reflects longstanding pressure overload and ensuing right heart failure (RHF) [34]. Multiorgan involvement and ultimately RHF play important pathophysiological roles [34]. However, the relationship between RV size and survival was active in the present research. The stiffness of a hypertrophied RV is a predisposing factor for arrhythmia, such as ventricular or supraventricular arrhythmia, leading to rapid and fatal haemodynamic damage and sudden death in patients with severe PAH [35].

To predict severe CLD-PH, we created a comprehensive echocardiographic scoring index combining PASP and additional PH signs [15], which displayed a satisfactory ability to predict severe PH and was recommended for application because of its cost-effectiveness and non-invasive nature.

The severity of PH should be judged in terms of hard endpoints, such as mortality, not just by pulmonary artery pressure (PAP). Unlike our previous diagnostic study, the present study strengthened the predictive nature of the composite scores of echocardiography regarding hard endpoints.

Despite the availability of PAH-targeted drugs, no such drugs are approved for CLD-PH, and their usage would be considered off-label in such populations. As a PH screening tool, PASP can be recommended for early screening and assessment of the prognosis of patients with IPAH, CTD-PAH or chronic thromboembolic PH [36]. Nevertheless, in patients with CLD, PASP estimation is often inaccurate and requires optimal visualization of the regurgitant jet and the presence of sufficient tricuspid regurgitation (TR). Moreover, the TR velocity might be underestimated in CLD patients due to marked respiratory variations and hyperinflation of the lungs [37]. At present, the ESC guidelines recommend grading the probability of PH based on TR velocity and other prespecified echocardiographic variables that suggest PH [13]. This raises the question of whether echocardiography can be applied to identify a high-risk population that could benefit from aggressive treatment, such as lung transplantation. In this study, we derived a comprehensive score combining RVEDTD, TAPSE, and PASP, and this score had a high AUC (C-statistic from 0.591 to 0.746) on ROC analysis.

In our analysis of the COPD subgroup, due to the small population, only RVEDTD was associated with all-cause death by multivariate Cox regression. This result implied that right ventricular remodelling predicted long-term adverse events. Echocardiography can evaluate right ventricular remodelling, including size and function.

From another point of view, our data may not be suitable for subgroup analysis because of the relatively small number of patients included in this study, although it would have been interesting to provide survival for each category, such as COPD, interstitial lung disease (ILD) and pneumoconiosis. Another important reason is that many patients with CLD usually have multiple causes and risk factors in the real world. Because of the complexity of the disease, it is sometimes very difficult to determine the accuracy category. Studying patients with CLD as a whole population in the real world may have more clinical significance. We will continue to accumulate more patients and provide survival data for each category in future studies.

### Study Limitations

First, this study may have patient selection bias because of its single-centre retrospective nature. Nevertheless, patients admitted to our centre come from all over China, which may reduce bias. It is necessary to design a new multicentre study for further evaluation and verification of the results. Second, the treatment of the underlying CLD or PAH-targeted therapies were not adjusted in this analysis due to heterogeneity. However, the included patients received similar levels of therapy and support.

## 5. Conclusions

Based on comprehensive echocardiography, the risk score can be used in routine follow-up of patients with severe CLD-PH. Our study supports the use of echocardiographic parameters, including PASP, RVEDTD and TAPSE, for predicting severe CLD-PH patient outcomes regarding all-cause mortality.

## Figures and Tables

**Figure 1 jcm-11-01603-f001:**
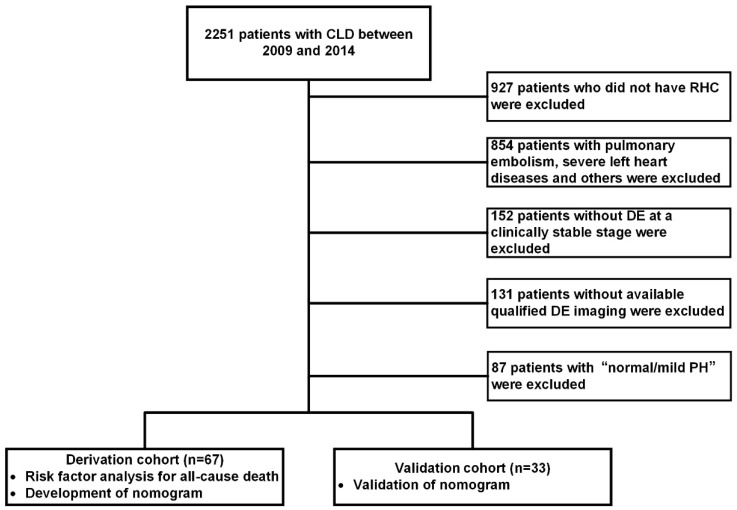
Flow diagram of patient enrolment and study design. CLD: chronic lung diseases; DE: Doppler echocardiography; RHC: right heart catheterization.

**Figure 2 jcm-11-01603-f002:**
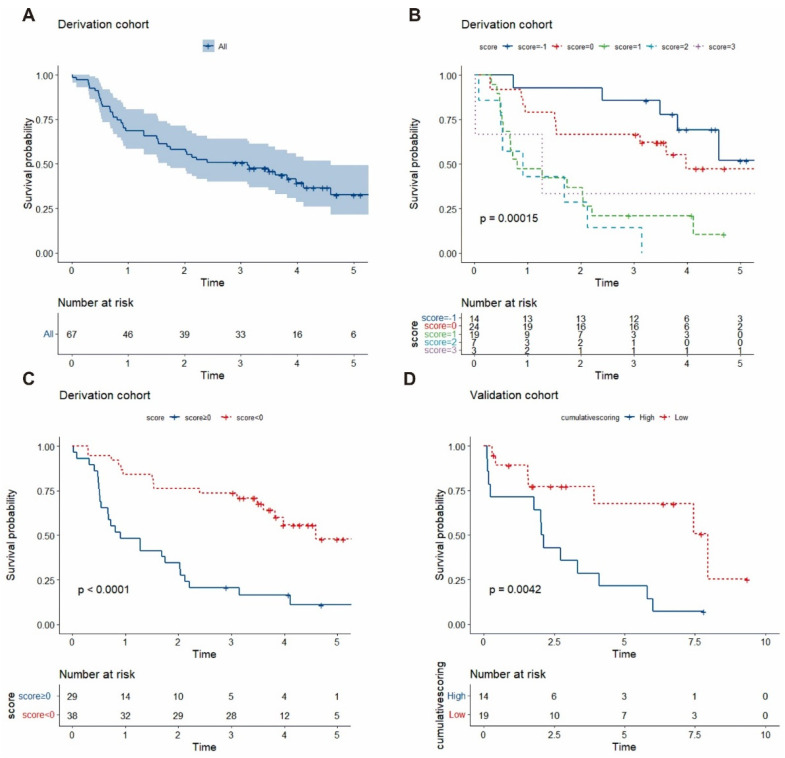
Kaplan–Meier survival for all-cause mortality of severe CLD-PH: (**A**) in the derivation cohort population; (**B**) in subgroups with scores of −1, 0, 1, 2 and 3; (**C**) in subgroups with scores < 0 vs. scores ≥ 0; (**D**) in the validation cohort according to the mean value of the total points calculated by each patient’s nomogram.

**Figure 3 jcm-11-01603-f003:**
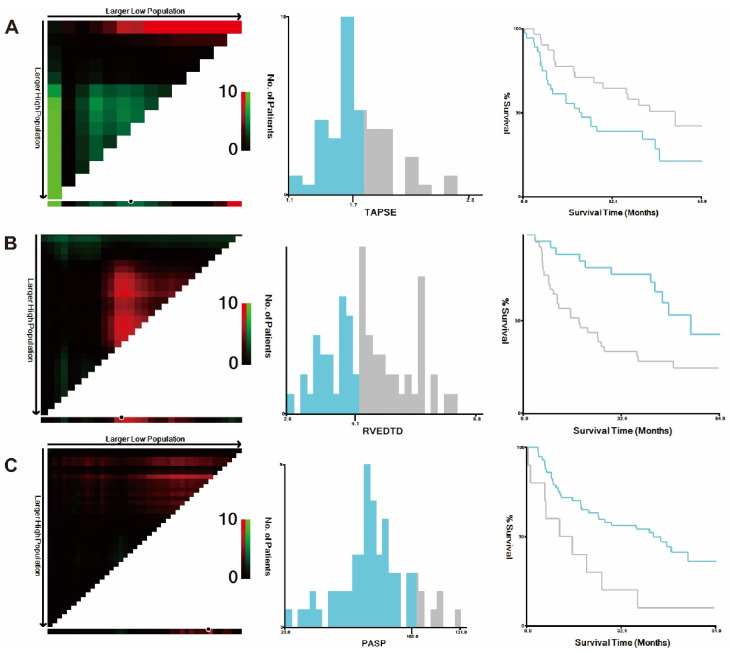
Cut-off values for echocardiographic parameters calculated using the X-tile program. X-tile analyses of TAPSE (**A**), PASP (**B**) and RVEDTD (**C**) levels in the cohort population with severe CLD-PH. X-tile plots for the cohort patients are shown in the left panels; black circles highlight the cut-off values, which are also shown in histograms (middle panels). Kaplan–Meier plots are presented in the right panels. RVEDTD: right ventricular end-diastolic transverse dimension; PASP: pulmonary arterial systolic pressure; TAPSE: tricuspid annular plane systolic excursion.

**Figure 4 jcm-11-01603-f004:**
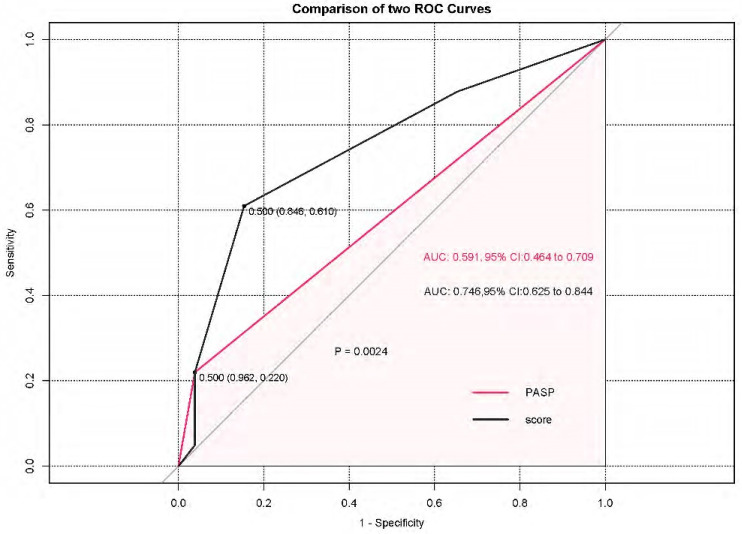
Comparisons of ROC curves of PASP alone and the composite scores in predicting all-cause mortality in patients with severe CLD-PH. AUC: area under the curve; PASP: pulmonary arterial systolic pressure; CLD: chronic lung diseases; PH: pulmonary hypertension; ROC: receiver operating characteristic.

**Figure 5 jcm-11-01603-f005:**
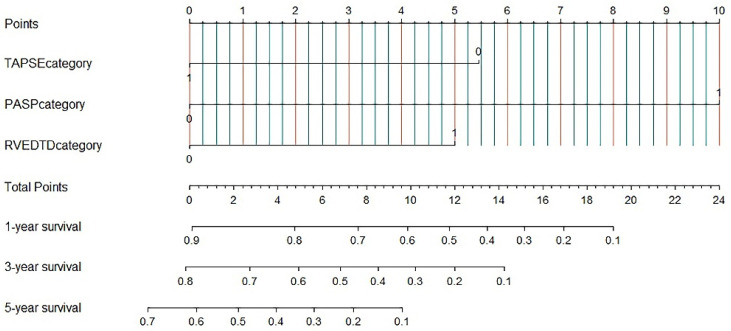
Nomogram for predicting all-cause mortality in severe CLD-PH. PASP: pulmonary arterial systolic pressure; RVEDTD: right ventricular end-diastolic transverse dimension; TAPSE: tricuspid annular plane systolic excursion.

**Figure 6 jcm-11-01603-f006:**
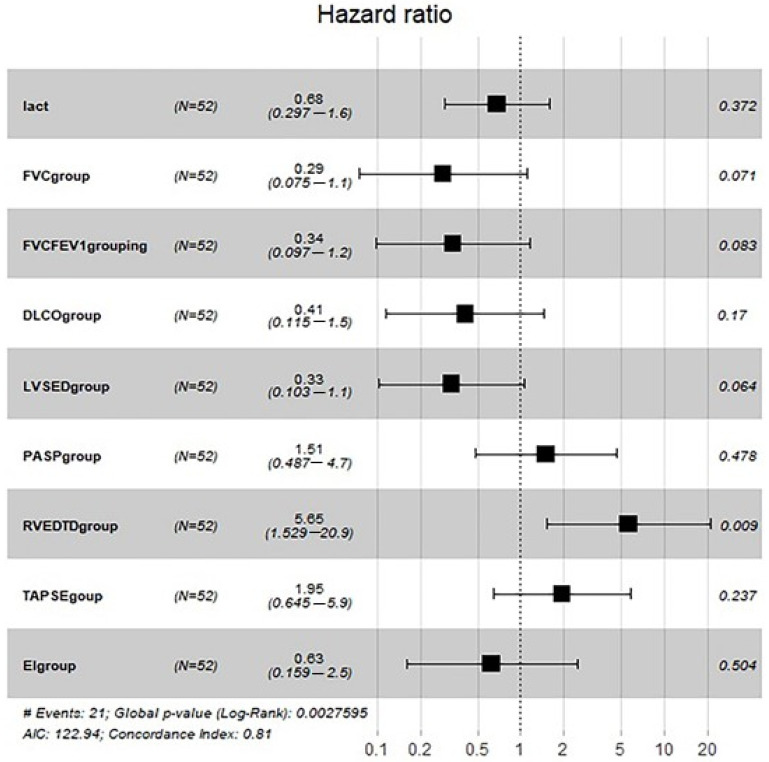
Cox univariate regression analyses for all-cause mortality among COPD patients. COPD: chronic obstructive pulmonary diseases; FEV1: forced expiratory volume in 1 s; FVC: forced vital capacity; DLco: diffusing capacity for carbon monoxide; RVEDTD: right ventricular end-diastolic transverse dimension; PASP: pulmonary arterial systolic pressure; ENDSEI: end-systolic stage eccentricity index; TAPSE: tricuspid annular plane systolic excursion.

**Table 1 jcm-11-01603-t001:** Demographics and Baseline Characteristics of Severe PH in the Derivation and Validation Cohorts.

Variable	Derivation Cohort	Validation Cohort	*p* Value
	(*n* = 67)	(*n* = 33)	
Age (years)	58.0 ± 12.9	61.8 ± 11.3	0.142
Male, *n* (%)	38 (56.7)	24 (84.8)	0.187
BSA (m^2^)	1.60 ± 0.2	1.59 ± 0.1	0.797
Aetiology, *n* (%)			
COPD	52 (77.6)	24 (72.4)	0.490
Interstitial lung disease	8 (11.9)	2 (6.1)	0.339
Asthma	3 (4.5)	0	0.549
Pneumoconiosis	2 (3.0)	1 (3.0)	1.000
CPFE	3 (4.5)	1 (3.0)	0.928
Post tubercular sequelae	1 (1.5)	0	1.000
Sleep-disordered breathing	2 (3.0)	1 (3.0)	1.000
Chest wall abnormalities	1 (1.5)	0	1.000
Comorbidities, *n* (%)			
Systemic hypertension	10 (14.9)	4 (12.1)	1.000
Diabetes mellitus	2 (3.0)	1 (3.0)	1.000
Coronary heart disease	1 (1.5)	1 (3.0)	1.000
Malignant tumour	1 (1.5)	0	1.000
Arrhythmia	3 (4.5)	2 (6.1)	1.000
Hyperlipidaemia	0	1 (3.0)	0.330
Previous cerebral infarction	1 (1.5)	0	1.000
Pulmonary function test			
FEV1% predicted	32.3 (24.0, 44.0)	35.5 (35.6, 49.4)	0.127
FVC% predicted	54.9 (40.9, 63.4)	60.1 (57.9, 72.7)	0.081
FEV1/FVC	47.4 (42.9, 56.1)	47.7 (47.1, 56.1)	0.823
RV% predicted	197.9 (137.9, 251.4)	116.4 (155.7, 206.3)	0.220
TLC% predicted	115.5 (92.0, 131.7)	108.8 (98.7, 117.6)	0.440
DLco% predicted	40.9 (27.9, 56.2)	40.9 (36.4, 61.6)	1.000
Haemodynamics			
mRAP, mmHg	7.0 (4.8, 9.3)	7.0 (5.6, 8.5)	0.541
mPAP, mm Hg	46.0 (42.0, 55.0)	44.0 (41.8, 49.2)	0.108
PAWP, mm Hg	9.0 (6.0, 13.0)	8.0 (7.6, 10.2)	0.348
CO, L/min	4.6 (4.0, 5.7)	4.5 (4.1, 5.3)	0.139
CI, L/min/m^2^	3.1 (2.6, 3.7)	2.7 (2.5, 3.1)	0.035
PVR, Wood units	8.0 (6.3, 10.2)	8.0 (7.1, 9.1)	0.895
Echocardiography			
LVEF, %	70.9 ± 9.0	68.9 ± 9.9	0.315
LA, cm	3.1 ± 0.6	3.5 ± 0.7	0.799
LVDED, cm	4.0 (3.5, 4.4)	3.9 (3.7, 4.2)	0.629
LVSED, cm	2.3 ± 0.6	2.4 ± 0.6	0.949
RATD, cm	4.8 (4.3, 5.5)	4.9 (4.8, 5.6)	0.200
RALD, cm	5.2 (4.3, 5.9)	5.0 (5.4, 5.9)	0.175
RVEDTD, cm	4.3 (3.8, 5.0)	4.3 (4.0, 4.9)	0.670
RVEDLD, cm	6.6 ± 0.9	6.8 ± 0.9	0.271
PASP, mmHg	76.2 ± 22.6	74.1 ± 24.4	0.533
TAPSE, cm	1.7 (1.5, 1.9)	1.8 (1.6, 1.8)	0.921
PAd, cm	2.9 (2.6, 3.2)	3.1 (2.9, 3.4)	0.208
ENDSEI	1.3 (1.0, 1.5)	1.1 (1.1, 1.3)	0.158
TV s’, cm/s	12.0 (9.0, 13.0)	11.2 (10.1, 11.6)	0.608
PASP/TAPSE, mmHg/cm	44.3 ± 14.4	44.4 ± 16.3	0.975
Blood gas analysis			
pH	7.39 (7.35, 7.42)	7.40 (7.38, 7.42)	0.189
PaO_2_, mmHg	58 (46, 60)	59.5 (50.8, 69.5)	0.591
PaCO_2_, mmHg	50.3 (39.7, 60.1)	46.5 (37.7, 58.6)	0.236
SaO_2_, %	87.0 (79.3, 91.7)	93.8 (83.1, 93.8)	0.136
PAH-targeted therapy			
PDE5I	47 (70.1)	30 (90.1)	0.127
ERA	5 (7.5)	1 (3.3)	0.661
Prostacyclin	6 (9.0)	1 (3.3)	0.420
None	9 (13.4)	2 (6.1)	0.330
Traditional treatment			
Oxygen	67 (100)	33 (100)	1.000
ICS/LABA	59 (88.1)	30 (90.9)	1.000

Values are expressed as the mean ± SD or median (quartile range). mPAP: mean pulmonary artery pressure; RAP: right atrial pressure; PAWP: pulmonary artery wedge pressure; CO: cardiac output; CI: cardiac index; PVR: pulmonary vascular resistance; COPD: chronic obstructive pulmonary disease; CPFE: combined pulmonary fibrosis and emphysema; PH: pulmonary hypertension; FEV1: forced expiratory volume in 1 s; FVC: forced vital capacity; RV: residual volume; TLC: total lung capacity; DLco: diffusing capacity for carbon monoxide; RVEDTD: right ventricular end-diastolic transverse dimension; RVEDLD: right ventricular end-diastolic longitudinal dimension; RATD: right atrial transverse dimension; RALD: right atrial longitudinal dimension; PASP: pulmonary arterial systolic pressure; ENDSEI: end-systolic stage eccentricity index; PAd: pulmonary artery dimension; TAPSE: tricuspid annular plane systolic excursion; LVEF: left ventricular ejection fraction; LVDED: left ventricular end-diastolic transverse dimension; LVSED: left ventricular end-systolic transverse dimension; TV s’: tricuspid myocardial systolic velocity; PaO_2_: oxygen partial pressure; PaCO_2_: partial pressure of carbon dioxide: SaO_2_: oxygen saturation; ICS/LABA, inhaled corticosteroids/long-acting β2-agonists; PDE5I: phosphodiesterase-5 inhibitors; ERA: endothelin receptor antagonist; PDE5I included sildenafil, vardenafil and tadalafil; ERA indicated bosentan; Prostacyclins included beraprost and inhaled iloprost.

**Table 2 jcm-11-01603-t002:** Cox univariate and multivariate regression analyses for all-cause mortality in the derivation cohort.

Variable *	Univariate Analysis		Multivariate Analysis			
	HR [95% CI]	*p* Value	HR [95% CI]	*p* Value	β-Coefficient	Weighted Scores
Age, years	1.581 (0.779, 3.206)	0.204				
Sex	0.691 (0.370, 1.290)	0.246				
BSA	1.004 (0.263, 3.833)	1.004				
PaO_2_, mm Hg	0.983 (0.963, 1.004)	0.119				
PaCO_2_, mm Hg	1.010 (0.984, 1.036)	0.471				
SaO_2_, %	0.973 (0.940, 1.008)	0.125				
Lactic acid, mmol/L	1.355 (0.891, 2.062)	0.155				
Echocardiography						
LVEF, %	0.531 (0.223, 1.268)	0.154				
LA, cm	0.666 (0.350, 1.265)	0.214				
LVDED, cm	0.608 (0.310, 1.193)	0.148				
LVSED, cm	0.408 (0.220, 0.756)	0.004				
RATD, cm	0.485 (0.201, 1.168)	0.107				
RALD, cm	1.588 (0.809, 3.118)	0.179				
RVEDTD, cm	2.583 (1.311, 5.091)	0.006	2.248 (1.062, 4.759)	0.034	0.810	1
RVEDLD, cm	1.363 (0.738, 2.519)	0.322				
PASP, mmHg	2.599 (1.232, 5.485)	0.012	5.039 (2.002, 12.680)	0.001	1.617	2
TAPSE, cm	0.514 (0.273, 0.969)	0.040	0.414 (0.196, 0.873)	0.021	−0.882	−1
PAd, cm	0.537 (0.253, 1.295)	0.181				
ENDSEI	2.342 (1.071, 5.123)	0.033				
Pulmonary function test						
FEV1% pred	0.753 (0.390, 1.455)	0.399				
FVC% pred	0.674 (0.362, 1.257)	0.215				
FEV1/FVC	0.427 (0.202, 0.904)	0.026				
RV% pred	1.197 (0.615, 2.330)	0.597				
TLC% pred	1.354 (0.660, 2.780)	0.409				
DLco% pred	0.437 (0.217, 0.879)	0.020				

* Data in parentheses are 95% confidence intervals (CIs). * Variables were transformed into binary variables according to optimum cut-off values. Abbreviations as in Table 1. If PASP ≥ 103 mm Hg, the weighted score = 2; If RVEDTD ≥ 4.2 cm, the weighted score = 1; If TAPSE ≥ 1.8 cm, the weighted score = −1.

**Table 3 jcm-11-01603-t003:** Example of score and point allocation.

Variables	Value
TAPSE ≥ 1.8 cm	−1
LVSED < 2.3 cm	0
RVEDTD ≥ 4.2 cm	+1
RVEDTD < 4.2 cm	0
PASP ≥ 103 mm Hg	+2
PASP < 103 mm Hg	0
Total Scores	−1~3

RVEDTD: right ventricular end-diastolic transverse dimension; PASP: pulmonary arterial systolic pressure; LVSED: left ventricular end-systolic transverse dimension; CI: cardiac index; TAPSE: tricuspid annular plane systolic excursion.

**Table 4 jcm-11-01603-t004:** Cox univariate regression analyses for all-cause mortality among COPD patients in the derivation cohort.

Variable *	HR [95% CI]	*p* Value
Lactic acid, mmol/L	2.877 (1.103, 7.503)	0.031
Echocardiography		
LVSED, cm	0.460 (0.231, 0.915)	0.027
RVEDTD, cm	3.708 (1.156, 9.069)	0.004
PASP, mmHg	1.432 (0.717, 2.858)	0.309
TAPSE, cm	2.040 (0.986, 4.218)	0.055
ENDSEI	2.267 (1.121, 4.587)	0.023
Pulmonary function test		
FVC% pred	0.467 (0.226, 0.966)	0.040
FEV1/FVC	0.473 (0.224, 0.997)	0.049
DLco% pred	0.260 (0.100, 0.676)	0.006

* Data in parentheses are 95% confidence intervals (CIs). * Variables were transformed into binary variables according to optimum cut-off values. Abbreviations as in Table 1.

## Data Availability

The datasets used and/or analysed during the current study are available from the corresponding author on reasonable request.

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
