# Peer review of "Echocardiography Nomogram for Predicting Survival among Chronic Lung Disease Patients with Severe Pulmonary Hypertension"

_jcm, 2022, doi:10.3390/jcm11061603_

Round 1
Reviewer 1 Report
Dear Author,
Post tubercular Sequelae also comes under chronic Lung Diseases, Kindly Specify PH in Post TB in your study.
As per the new definition of pulmonary hypertension, (PH) in between 20-25 mmHg would affect the study result Please specify.
Author Response
Point 1:Post tubercular Sequelae also comes under chronic Lung Diseases, Kindly Specify PH in Post TB in your study.
Response:Thanks for your important suggestion. “Sequelae of pulmonary tuberculosis” was replaced by “post tubercular sequelae”, and delete the sentence “For example, a CLD patient has COPD and sequelae of pulmonary tuberculosis”.
Point 2:As per the new definition of pulmonary hypertension, (PH) in between 20-25 mmHg would affect the study result Please specify.
Response: Thank very much for your suggestion. Although guidelines recommend that mPAP between 20 and 24mmHg be included in PH. However, the guideline does not change the concept of hemodynamic subset of CLD-PH. In addition, the study population in our study was CLD patients with severe PH, namely, mPAP ≥ 35 mmHg or 25 mmHg ≤ mPAP < 35 mmHg with CI < 2.0 L /min/m2 or PVR > 6 Wood units). So the new definition of PH would not affect the result of the study.
Reviewer 2 Report
This is a study on the prognosis of chronic lung disease patients with severe pulmonary hypertension with an interesting concept.
However it has some major flaws/limitations:
- The authors must state clearly in the first paragraph of the Methods that this is a retrospective analysis.
- The patients state that they reviewed consecutive patients with suspected CLD-associated PH and that all consecutive patients underwent Doppler echocardiography and RHC. However, the flowchart shows that out of the 2251 patients with suspected CLD-associated PH, 927 did not have RHC and another 131 did not have available qualified echo images. Therefore, the final sample is not comprised of consecutive patients and also given the large percentage of patients that were not included in the study a selection bias may be introduced. The authors should at least provide a descriptive table of patients included in the study and those who could potentially have severe PH but were excluded.
- There are large differences in the FEV1 and FVC variables between the two groups.
- The authors mention that they found optimal cut-off points for continuous parameters. However, this is not good statistical practice. Why did not they use the beta coefficient of the continuous parameters in the score? Or at least use cut off points that have been validated in previous literature.
- It is not clear what is the rationale behind testing RHC and LFT parameters in univariate analysis, but then not introducing the parameters with p<0.1 into the multivariable Cox model. The authors stress that this is a study that wants to focus on echocardiography, however introducing other parameters into the multivariable would change the beta coefficients and the whole score. Otherwise, there is no use of the univariate testing of RHC and LFT parameters.
- The subgroup analysis of COPD patients seems a bit out of the blue. Also, because there the authors used also other than echo parameters in the multivariable Cox…
Author Response
Point 1: The authors must state clearly in the first paragraph of the Methods that this is a retrospective analysis.
Response: Thanks for your important suggestion. We have stated the retrospective study in the first paragraph of the Methods.
Point 2: The patients state that they reviewed consecutive patients with suspected CLD-associated PH and that all consecutive patients underwent Doppler echocardiography and RHC. However, the flowchart shows that out of the 2251 patients with suspected CLD-associated PH, 927 did not have RHC and another 131 did not have available qualified echo images. Therefore, the final sample is not comprised of consecutive patients and also given the large percentage of patients that were not included in the study a selection bias may be introduced. The authors should at least provide a descriptive table of patients included in the study and those who could potentially have severe PH but were excluded.
Response: Thanks for your very important comment. As you said, this was a retrospective study. According to your suggestion, we have clearly stated this retrospective study in the article and deleted the ambiguous words. To avoid ambiguity, “2251 patients with suspected CLD-associated PH” was replaced by “2251 patients with CLD”. The corresponding figure was also modified accordingly. In addition, the “severe PH group” includes only a minority of CLD patients suspected of having significant/severe vascular abnormalities (remodeling) accompanying the parenchymal disease (Pulmonary hypertension due to chronic lung disease: Updated Recommendations of the Cologne Consensus Conference 2011). Selection bias was hard to avoid. “Severe CLD-PH” must be diagnosed by RHC, and we analyzed 1137 patients who undergone RHC but were excluded in the study. We briefly collected the basic information of these patients. The age was about 57.0 ± 12.8 years, with female of 41.5 %. The mPAP was about 38.7 ± 13.2mmHg, and CI was 3.4 ± 1.0 L/min/m2.
Point 3: There are large differences in the FEV1 and FVC variables between the two groups.
Response:We are sorry that we made a mistake in the order of FEV1% predicted and FVC % predicted. Now we have changed the order of the variable.
Point 4: The authors mention that they found optimal cut-off points for continuous parameters. However, this is not good statistical practice. Why did not they use the beta coefficient of the continuous parameters in the score? Or at least use cut off points that have been validated in previous literature.
Response:Thanks for your very important suggestion. (1) In our study, considering the time-dependent factor, we found the "optimal" cutoff value of the continuous variable by using x-tile software. We referred to the similar studies (PMID: 15534099). (2) We maintained the consistency of data and divided continuous variables into binary variables, making clinical practice more convenient and simpler. (3) Initially, we have used the cutoff values validated in previous literature, but the results were inferior to our cutoff values which were found by the “time dependence” method. In addition, in clinical practice, we have found echocardiography heterogeneity across different PH types. For example, compared with IPAH patients, those with congenital heart disease without repaired have significantly higher TAPSE and less dilated right heart. However, in fact, severe right heart remodeling has occurred. Due to the influence of lung diseases on acoustic window, the estimate of PASP may be less accurate. We believe that echocardiographic analysis with “disease characteristics” may be more meaningful. One advantage of our study is that the influence of time was taken into account when selecting the cutoff value. (4) We considered "optimal" to cause your ambiguity. Based on echocardiographic heterogeneity in different PH types, we dropped the “optimal” word after consulting with the statisticians. The “optimal cutoff” was replaced by “the cutoff”.
Point 5: It is not clear what is the rationale behind testing RHC and LFT parameters in univariate analysis, but then not introducing the parameters with p<0.1 into the multivariable Cox model. The authors stress that this is a study that wants to focus on echocardiography, however introducing other parameters into the multivariable would change the beta coefficients and the whole score. Otherwise, there is no use of the univariate testing of RHC and LFT parameters.
Response: Thanks for your very important comment. RHC is an invasive method, however, the number of patients with chronic lung disease is high and invasive right heart catheterization is not available in all hospitals of China. Noninvasive methods that can assess disease severity and prognosis may be more suitable for clinical use, especially in China. So, the aim of our study was to explore the value of non-invasive method, echocardiography in predicting the prognosis because of its convenience and other advantages. However, for chronic lung diseases, the results are difficult to interpret without an adjustment with variables such as lung functional data. Although lung function data was statistically significant in univariate regression, it was not in multivariate regression. With an adjustment of lung function correction, the results may be more reliable.
Point 6: The subgroup analysis of COPD patients seems a bit out of the blue. Also, because there the authors used also other than echo parameters in the multivariable Cox…
Response:Thanks for your very important comment. At the beginning, we analyzed the whole population with chronic lung disease and found that the three echocardiographic parameters, namely RVEDTD, PASP, TAPSE, had predictive value for the prognosis of the disease. However, when we performed a subgroup analysis of the COPD population, only RVEDTD was found to be significant. We think there may be a sample size problem, and maybe the number of COPD patients is large enough that the meaning of other parameters may be revealed. In addition, we found that echocardiography still predicted all-cause death of patient with severe CLD-PH even in subgroup analysis.
Reviewer 3 Report
The paper by Rong Jiang et al proposes an echocardiography nomogram for predicting survival among Chronic Lung Disease Patients with Severe Pulmonary Hypertension.
Major comments:
- Did patients have a RV dysfunction also in terms of fractional area change? Was RV hypertrophy present?
- Was there a drug effect?
- Was any advanced echocardiographic technique available? Such as RV free wall longitudinal strain and or three-dimensional RV echo?
- The composite score was assessed attributing one point to RVEDTD ≥ 4.2 cm, one point to PASP ≥ 103 mm Hg, and minus one point was attributed to TAPSE ≥ 1.8 cm. The cut-offs values for RVEDTD and TAPSE correspond to the thresholds of normalcy according to Lang R et al. Eur Heart J Cardiovasc Imaging. 2015 Mar;16(3):233-70. Thus, the score suggests that patients with RV dilation and RV dysfunction together with elevated PAPS has an increased risk of death at 5 years. How this add meaningful significance to current literature? What are the perspectives and the clinical implications of the present study? How the present nomogram could be useful into clinical practice?
Author Response
Point 1: Did patients have a RV dysfunction also in terms of fractional area change? Was RV hypertrophy present?
Response: Thanks for your important suggestion. We used TAPSE and TV S’ not the fractional area changes to reflect RV function. In the future study, we will apply FAC. We had measured right ventricular wall thickness (RVWT) as RV hypertrophy parameter. However, RVWT values varied 6mm-8mm (normal< 5mm) in our study population, and because there did not exist difference, it was not included in the study.
Point 2: Was there a drug effect?
Response: Thanks for your important suggestion. On the one hand, there was no difference in drugs between the two groups at baseline. This may suggest that the principle of drug treatment for patients in our center is unified. On the other hand, it is difficult for us to track the detailed changes of drugs for up to 7 years, such as whether interrupted? Interruption time? Has the variety of ICs drugs been changed? Two or three? Any changes in the types of PAH-targeted drugs or traditional drugs? Although drugs had impacts on the prognosis, it was difficult to analyze accurately because of complexity.
Point 3 Was any advanced echocardiographic technique available? Such as RV free wall longitudinal strain and or three-dimensional RV echo?
Response: Thanks for your constructive suggestion. Our research was based on the retrospective analysis of previous data. We did not carry out advanced Echo technique detection. We have planned to carry out longitudinal strain and 3 D technique in future. In addition, ordinary and conventional Echo parameters may have promotion advantage in township hospitals of China.
Point 3: The composite score was assessed attributing one point to RVEDTD ≥ 4.2 cm, one point to PASP ≥ 103 mm Hg, and minus one point was attributed to TAPSE ≥ 1.8 cm. The cut-offs values for RVEDTD and TAPSE correspond to the thresholds of normalcy according to Lang R et al. Eur Heart J Cardiovasc Imaging. 2015 Mar;16(3):233-70. Thus, the score suggests that patients with RV dilation and RV dysfunction together with elevated PAPS has an increased risk of death at 5 years. How this add meaningful significance to current literature? What are the perspectives and the clinical implications of the present study? How the present nomogram could be useful into clinical practice?
Response: Thanks for your very important suggestion. RHC is the gold standard for the diagnosis of severe PH because this method provides the hemodynamic information that defines this disease. Nevertheless, RHC is not routinely and repeatedly performed at initial diagnosis of PH and follow-up, especially in China, limiting where performance of RHC is limited by its invasiveness and high expenses. As a PH screening tool, systolic PAP (PASP) can be estimated by measuring the peak tricuspid regurgitation velocity on echocardiography, which continues to be recommended for early screening and as an assessment tool in patients with PAH. However, in spite of its widespread use, the accuracy and reproducibility of echocardiography in predicting PASP have recently been questioned1-6.
The updated European Society of Cardiology and European Respiratory Society guidelines on PH recommend testing for additional PH signs by assessing pulmonary artery diameter (PAd) and right ventricle (RV) enlargement in addition to PASP7. In our previous study, we established a novel scoring index with guideline-recommended echocardiographic parameters to predict severe CLD-PH8, which showed high capacity for predicting severe CLD-PH, further implying the value of noninvasive examinations in clinic. Early diagnosis and treatment are crucial in the patient population. Compared with identifying severe CLD-PH, predicting the prognosis of this population is of more practical significance. So, we conducted a new simply noninvasive comprehensive echocardiographic index to predict the prognosis for early identify “high risk” patient and timely and adequate intervention. The nomogram is more intuitive to predict the prognosis.
- Robinson B, Ebeid M. A simple echocardiographic method to estimate pulmonary vascular resistance. The American journal of cardiology. 2014;113:412
- Opotowsky AR, Clair M, Afilalo J, Landzberg MJ, Waxman AB, Moko L, Maron BA, Vaidya A, Forfia PR. A simple echocardiographic method to estimate pulmonary vascular resistance. The American journal of cardiology. 2013;112:873-882
- D'Alto M, Romeo E, Argiento P, D'Andrea A, Vanderpool R, Correra A, Bossone E, Sarubbi B, Calabro R, Russo MG, Naeije R. Accuracy and precision of echocardiography versus right heart catheterization for the assessment of pulmonary hypertension. International journal of cardiology. 2013;168:4058-4062
- Aduen JF, Castello R, Daniels JT, Diaz JA, Safford RE, Heckman MG, Crook JE, Burger CD. Accuracy and precision of three echocardiographic methods for estimating mean pulmonary artery pressure. Chest. 2011;139:347-352
- Roule V, Labombarda F, Pellissier A, Sabatier R, Lognone T, Gomes S, Bergot E, Milliez P, Grollier G, Saloux E. Echocardiographic assessment of pulmonary vascular resistance in pulmonary arterial hypertension. Cardiovascular ultrasound. 2010;8:21
- Arcasoy SM, Christie JD, Ferrari VA, Sutton MS, Zisman DA, Blumenthal NP, Pochettino A, Kotloff RM. Echocardiographic assessment of pulmonary hypertension in patients with advanced lung disease. American journal of respiratory and critical care medicine. 2003;167:735-740
- Galie N, Humbert M, Vachiery JL, Gibbs S, Lang I, Torbicki A, Simonneau G, Peacock A, Vonk Noordegraaf A, Beghetti M, Ghofrani A, Gomez Sanchez MA, Hansmann G, Klepetko W, Lancellotti P, Matucci M, McDonagh T, Pierard LA, Trindade PT, Zompatori M, Hoeper M, Aboyans V, Vaz Carneiro A, Achenbach S, Agewall S, Allanore Y, Asteggiano R, Paolo Badano L, Albert Barbera J, Bouvaist H, Bueno H, Byrne RA, Carerj S, Castro G, Erol C, Falk V, Funck-Brentano C, Gorenflo M, Granton J, Iung B, Kiely DG, Kirchhof P, Kjellstrom B, Landmesser U, Lekakis J, Lionis C, Lip GY, Orfanos SE, Park MH, Piepoli MF, Ponikowski P, Revel MP, Rigau D, Rosenkranz S, Voller H, Luis Zamorano J. 2015 esc/ers guidelines for the diagnosis and treatment of pulmonary hypertension: The joint task force for the diagnosis and treatment of pulmonary hypertension of the european society of cardiology (esc) and the european respiratory society (ers): Endorsed by: Association for european paediatric and congenital cardiology (aepc), international society for heart and lung transplantation (ishlt). Eur Heart J. 2016;37:67-119
- Jiang R, Wu C, Pudasaini B, Wang L, Zhao QH, Zhang R, Wu WH, Yuan P, Jing ZC, Liu JM. A novel scoring index by doppler echocardiography for predicting severe pulmonary hypertension due to chronic lung diseases: A cross-sectional diagnostic accuracy study. International journal of chronic obstructive pulmonary disease. 2017;12:1741-1751
Round 2
Reviewer 2 Report
The authors did not address my recommendations adequately.
Reviewer 3 Report
No further comments
This manuscript is a resubmission of an earlier submission. The following is a list of the peer review reports and author responses from that submission.